# Age and political leaning predict COVID-19 vaccination status at a large, multi-campus, public university in Pennsylvania: A cross-sectional survey

**Ryan Murphy**[1], **Lauren Pomerantz**[1], **Prabhani Kuruppumullage Don**[2], **Jun Sung Kim**[2], **Bradley A. Long**[1,3] *

**1** Pennsylvania State University College of Medicine, University Park Regional Campus, State College, Pennsylvania, United States of America, **2** Department of Statistics, Pennsylvania State University, University Park, Pennsylvania, United States of America, **3** Harrell Health Sciences Library, Pennsylvania State University College of Medicine, Hershey, Pennsylvania, United States of America

* blong3@pennstatehealth.psu.edu

## Abstract

### Introduction

Vaccine hesitancy during the COVID-19 pandemic impacted many higher education institutions. Understanding the factors associated with vaccine hesitancy and uptake is instrumental in directing policies and disseminating reliable information during public health emergencies.

### Objective

This study evaluates associations between age, gender, and political leaning in relationship to COVID-19 vaccination status among a large, multi-campus, public university in Pennsylvania.

### Methods

From October 5—November 30, 2021, a 10-minute REDCap survey was available to students, faculty, and staff 18 years of age and older at the Pennsylvania State University (PSU). Recruitment included targeted email, social media, digital advertisements, and university newspapers. 4,231 responses were received. Associations between the selected factors and vaccine hesitancy were made with Chi-square tests and generalized linear regression models using R version 4.3.1 (2023-06-16).

### Results

Logistic regression approach suggested that age and political leaning have a statistically significant association with vaccine hesitancy at the 5% level. Adjusted for political leaning, odds of being vaccinated is 4 times higher for those aged 56 years or older compared to the ones aged 18 to 20 (OR = 4.35, 95% CI = (2.82, 6.85), p-value < 0.05). The results also

**Data Availability Statement:** All relevant data are within the manuscript and its Supporting information files.

**Funding:** RM and LP received funding from the The Harrell Health Sciences Library at the Penn State College of Medicine to receive compensation as research assistants. The funders had no role in study design, data collection and analysis, decision to publish, or preparation of the manuscript.

**Competing interests:** The authors have declared that no competing interests exist.

showed that adjusted for age, the odds of being vaccinated is about 3 times higher for liberal individuals compared to far-left individuals (OR = 2.85, 95% CI = (1.45, 5.41), p-value = 0.001).

## Conclusions

Age and political leaning are key predictors of vaccine uptake among members of the PSU community, knowledge of which may inform campus leadership's public health efforts such as vaccine campaigns and policy decisions.

## Introduction

Vaccine hesitancy during the COVID-19 pandemic leads to suboptimal uptake and poses a concern to highly vulnerable populations to SARS-CoV-2 infection [1]. A preliminary national survey revealed that before COVID-19 vaccines were available, certain demographic factors predicted vaccine mandate support, such as age, political party affiliation, gender, race, and rurality [1–3]. After vaccines became available, the major predictors of vaccine hesitancy were further elucidated [2, 4, 5]. While many predictors exist, the 5C model helps organize these drivers of vaccine hesitancy based on five major factors: confidence, complacency, convenience or constraints on convenience, risk calculation and collective responsibility [6]. Specific examples of major predictors include being unvaccinated against influenza, having a lower perceived risk of infection/infection severity, believing that vaccines are unsafe, and maintaining lower institutional trust [2, 4, 5]. Political leaning has also been associated with vaccine hesitancy [2].

While there are global consequences of vaccine hesitancy, uncovering its impact at large, public universities in Pennsylvania have yet to be fully evaluated. The present study evaluated which demographic factors were associated with vaccine hesitancy and uptake at the Pennsylvania State University (PSU) in the fall of 2021, a period in which COVID-19 vaccines were widely available to the public. PSU is one of Pennsylvania's largest employers and boasts a diverse, global population [7]. With this understanding, there needs to be an emphasis on targeted vaccination and information campaigns during public health emergencies, rather than a blanket approach. Furthermore, two studies examined vaccination incentives, specifically lotteries, and how they may impact vaccine uptake. One study found, that overall, across 11 states, vaccination increased by approximately 23% with the initiation of a vaccine lottery program, though success differed greatly amongst states, reaffirming the need for heterogenous approaches to vaccine incentive programs [8]. Additionally, it was determined that vaccine lotteries are most effective when incentivizing booster doses when barriers to vaccination are greatest [9].

It should be noted that aspects of a university setting (e.g., social gatherings, having a roommate) may facilitate the spread of COVID-19, emphasizing the need to study this population [10, 11]. Additionally, comparing the views of different groups on campus, such as the student body (including undergraduate, graduate, and professional students), faculty, and staff may aid in the development of targeted policies that promote evidence-based measures such as vaccination [1, 12]. For context on the COVID-19 positivity rate on all PSU campuses during the study timeframe (October through November of 2021), weekly positivity rates ranged from 0.8–2.8% for employees, and 0.7–3.0% for students, with up to 12,000+ tests administered weekly [13].

## Methods/Subjects

This was a cross-sectional study that took place at all 23 PSU physical campuses, the World Campus, the Penn State Extension, and Penn State Health. Students, faculty, and staff 18 years of age and older were invited to take a REDCap survey between October 5 and November 30 of 2021. The Pennsylvania College of Technology was excluded, as it is only an affiliated institutional and not fully part of PSU. Survey questions were repeatedly tested by medical students, faculty, and staff prior to official launch to receive feedback about question clarity. Participants were recruited by randomly emailing 32,792 individuals with a PSU email address. Additionally, digital and print marketing campaigns took place with campus websites, bulletins, and newspapers. This study was carried out in accordance with the institutional review board at the Penn State College of Medicine (reference number 00016908), which deemed this project exempt. Digital, written consent to participate was obtained from participants. They were provided with details and instructions about the research project and asked three qualifying questions. The survey only continued if participants answered "Yes" to all three qualifying questions, which were: consenting to the survey's instructions, being 18 years of age or older, and being affiliated with the Pennsylvania State University during the study period. If a participant answered "No" to any of the criteria, the survey did not continue. The authors did not have access to information that could identify participants before, during or after data collection.

The REDCap survey incorporated both original research questions and validated questions from previous studies. Political leaning was measured by modifying the three-point Gallup item [14]. Rather than asking if participants are Republican, Democrat, or Independent, they were asked to identify with one of the following: Far left, Liberal, Middle-of-the-road, Conservative, Far right, or Prefer not to answer. The authors believe avoiding specific political parties and expanding the number of responses captures more granularity than the three-point Gallup item. Recruitment methods included targeted advertisements and messaging through university email, social media outlets, digital resources, and university newspapers.

Chi-square tests and generalized linear regression models on R version 4.3.1 (2023-06-16) were used to evaluate associations between vaccine hesitancy (Table 4) and selected survey factors (gender, age, and political leaning). These factors were chosen because prior studies showed they were important in determining vaccine hesitancy [1–3]. Chi-square tests suggested that age, gender, and political leaning were independently and statistically significantly associated with vaccine status at 10% level. However, gender, when adjusted for age and political leaning, became non-significant and was removed from the model. Vaccine hesitancy was determined by the following survey item: *"Have you been vaccinated for COVID-19 or are you planning to get vaccinated?"* and the answers "No" or "Undecided" signaled vaccine hesitancy (Table 4).

## Results

4,231 survey responses were received. 32,792 emails were randomly sent to those with a PSU email address, making the response rate 12.90%. After excluding participant responses that did not complete all questions of interest for this study, 3,145 cases were ultimately used for analysis. Most participants were between the ages of 26 and 35 (21.0%), women (57.2%), heterosexual (78.0%), and White (77.5%) (Table 1). Most respondents were employees (52.6%), and of the students who responded, most were undergraduates seeking a bachelor's degree (63.3%) (Table 2). Additionally, most considered their political views to be liberal (32.9%) (Table 3). Regarding prior COVID-19 infection and vaccination status, most never tested positive for COVID-19 (80.3%), have been vaccinated or plan to get vaccinated (80.9%), and receive the seasonal influenza vaccine (61.4%) (Table 4).

**Table 1. Participant age, gender, sexual orientation and race.**

| Measure | Response | Count | Percentage |
|---|---|---|---|
| Age | | | |
| | 18–20 | 695 | 17.2% |
| | 21–25 | 728 | 18.0% |
| | 26–35 | 851 | 21.0% |
| | 36–45 | 626 | 15.5% |
| | 46–55 | 575 | 14.2% |
| | 56–65+ or Other | 568 | 14.1% |
| Gender | | | |
| | Man | 1584 | 39.3% |
| | Woman | 2308 | 57.2% |
| | Other/Prefer not to answer | 143 | |
| Sexual Orientation | | | |
| | Straight | 3122 | 78.0% |
| | Gay | 107 | 2.7% |
| | Lesbian | 71 | 1.8% |
| | Other/Prefer not to answer | 705 | 17.5% |
| Race | | | |
| | Asian | 198 | 5.0% |
| | Black | 130 | 3.3% |
| | White | 3095 | 77.5% |
| | Other/Two or more/Prefer not to answer | 571 | 14.5% |

Age, political leaning, and gender were all significantly and independently associated with COVID-19 vaccine hesitancy status at 10% level (p< .001, p< .001, and p = .074, respectively). Logistic regression analysis for sexual orientation and race were not performed, since respondents were overwhelmingly White and heterosexual. Logistic regression approach suggested

**Table 2. Participant role at PSU, employee type and student level.**

| Measure | Response | Count | Percentage |
|---|---|---|---|
| Role at Penn State | | | |
| | Employee | 2035 | 52.6% |
| | Student | 1756 | 45.4% |
| | Prefer not to answer | 76 | 2.0% |
| Employee type | | | |
| | Administrator or Executive | 184 | 9.1% |
| | Staff (all categories) | 1400 | 69.5% |
| | Faculty | 367 | 18.2% |
| | Postdoc, fellow, or medical resident | 32 | 1.6% |
| | Prefer not to answer | 30 | 1.5% |
| Student level | | | |
| | Undergraduate, Bachelor's degree | 1103 | 63.3% |
| | Undergraduate, other | 180 | 10.4% |
| | Graduate, Master's degree | 231 | 13.3% |
| | Graduate, other | 215 | 12.3% |
| | Prefer not to answer | 13 | 0.7% |

**Table 3. Participant political views.**

| Measure | Response | Count | Percentage |
|---|---|---|---|
| Political views | | | |
| | Far left | 290 | 7.5% |
| | Liberal | 1271 | 32.9% |
| | Middle-of-the-road | 1141 | 29.6% |
| | Conservative | 766 | 19.8% |
| | Far right | 68 | 1.8% |
| | Prefer not to answer | 325 | 8.4% |

that gender, when adjusted for age and political leaning, does not have a statistically significant association with vaccine hesitancy status. Regarding age and the odds of reporting COVID-19 vaccination, odds increase with each increase in age group, when compared to the 18-20-year-old age group, after adjusting for political leaning. The highest odds were 4.35, observed in the 56 years of age or older group (Table 5), suggesting that when adjusted for political leaning, those older than 55 years of age are about 4 times likely to get vaccinated compared to 18-20-year-olds (95% CI for OR = (2.78, 6.68)). Regarding political leanings and the odds of reporting COVID-19 vaccination after adjusting for age, liberals' odds were the highest at 2.85, while far-right odds were the lowest at 0.06, when compared to the far-left (Table 6). That is, adjusted for age, odds of getting vaccinated was about 3 times higher for liberals compared to far-lefts (OR = 2.85, 95% CI for OR = (1.45, 5.41)) where as far-rights are 94% less likely to be vaccinated than far-lefts (OR = 0.06, 95% CI for OR = (0.027, 0.134)).

## Discussion

Using a REDCap survey distributed from October 5-November 30, 2021 to the PSU community, factors like age, political leanings, and gender were evaluated for their association with self-reported COVID-19 vaccination status. During the survey period, COVID-19 vaccines

**Table 4. History of a COVID-19-positive test and vaccine statuses for COVID-19 and influenza.**

| Measure | Response | Count | Percentage |
|---|---|---|---|
| Have you ever tested positive for COVID-19? | | | |
| | Yes | 694 | 18.2% |
| | No | 3055 | 80.3% |
| | Prefer not to answer | 57 | 1.5% |
| Have you been vaccinated for COVID-19 or are you planning to get vaccinated? | | | |
| | Yes | 2995 | 80.9% |
| | No | 520 | 14.0% |
| | Undecided | 126 | 3.4% |
| | Prefer not to answer | 63 | 1.7% |
| Do you receive the seasonal influenza vaccine when it becomes available each year?[a] | | | |
| | Yes | 2281 | 61.4% |
| | No | 793 | 21.3% |
| | Not always | 606 | 16.3% |
| | Prefer not to answer | 35 | 0.9% |

[a]Asking whether participants receive the seasonal influenza vaccine served as a marker of vaccine hesitancy [15].

**Table 5. Odds of reporting COVID-19 vaccination based on age group.**

| Age group (years) | Number vaccinated/total (%) | Estimated coefficient and standard deviation in logistic model[a] | Odds ratio of reporting vaccination, compared to 18-20-year-olds[a] (95% CI for OR) |
|---|---|---|---|
| Reference Level: 18–20 | 377/497 (75.85%) | N/A | N/A |
| 21–25 | 474/560 (84.64%) | 0.35 ± 0.176 | 1.42 (1.01, 2.01) |
| 26–35 | 586/662 (88.52%) | 0.37 ± 0.180 | 1.45 (1.02, 2.06) |
| 36–45 | 434/490 (88.57%) | 0.58 ± 0.195 | 1.79 (1.22, 2.63) |
| 46–55 | 423/481 (87.94%) | 0.80 ± 0.191 | 2.23 (1.54, 3.27) |
| 56+ | 422/455 (92.75%) | 1.47 ± 0.225 | 4.35 (2.82, 6.85) |

[a]Corrected for political leaning.

CI = confidence interval

OR = odds ratio

**Table 6. Odds of reporting COVID-19 vaccination based on political leanings.**

| Political leaning | Number vaccinated/total (%) | Estimated coefficient and standard deviation in logistic model[a] | Odds ratio of reporting vaccination, compared to the far-left[a] (95% CI for OR) |
|---|---|---|---|
| Reference Level: Far-left | 221/236 (93.64%) | N/A | N/A |
| Liberal | 1148/1174 (97.78%) | 1.05 ± 0.333 | 2.85 (1.45, 5.41) |
| Middle-of-the-road | 915/1035 (88.41%) | -0.77 ± 0.285 | 0.46 (0.25, 0.78) |
| Conservative | 412/656 (62.80%) | -2.28 ± 0.282 | 0.10 (0.06, 0.17) |
| Far-right | 20/44 (45.45%) | -2.79 ± 0.407 | 0.06 (0.03, 0.13) |

[a]Corrected for age.

CI = confidence interval

OR = odds ratio

were readily available, with the United States Food and Drug Administration approving the first vaccine on August 23, 2021, for individuals 16 years of age and older [16].

In total, approximately 81% of participants had either been vaccinated against COVID-19 or were planning to do so while only 14% stated they would not get vaccinated (Table 4). Further analysis revealed that those aged 56 years or older and those of liberal political leanings were more likely to report COVID-19 vaccination. Knowledge of this has the potential to influence leadership's approach to ensuring public health on campus. For example, communications primarily targeting the 56+ year old and/or liberal-leaning political groups may not be as fruitful as those targeting groups with lower vaccine uptake. Given the compelling evidence supporting vaccination to reduce mortality from COVID-19, it is in the best interest of PSU to strategize ways to increase vaccine uptake in such groups [17].

Future projects would involve piloting interventions, such as vaccine incentive programs, that target the groups with lower vaccination rates and assessing if vaccine uptake meaningfully increased within those groups. Given the compelling evidence supporting vaccination to reduce mortality from COVID-19, and the emergence of novel COVID-19 variants, it is in the best interest of PSU to strategize ways to increase vaccine uptake [17]. Although not trialed at PSU, vaccine lotteries have shown efficacy in increasing uptake, although with mixed success depending on geographic location (i.e., state) [8]. Booster dose uptake is more likely to be

influenced by vaccine lottery incentives than initial doses [9]. Both findings highlight the importance of utilizing heterogenous approaches to increasing vaccine uptake, considering both geographic factors as well as the vaccine dose type, when formulating incentives.

Regarding the higher vaccine uptake in those ages 56 years or older, it is plausible that younger participants, on average, had fewer risk factors for serious illness from COVID-19 infection and therefore may have perceived lower need or desire for vaccination, a finding supported elsewhere [18, 19]. It is important to consider that at the time of this survey, 80.3% of participants never knowingly had a positive COVID-19 test (Table 4), which may have influenced participants' perceived need to get a vaccine. It is also important to note the timing of when the survey was available, as participation took place prior to the highly contagious Omicron variant serving as the predominant variant starting in December 2021. It is plausible that the number of participants reporting prior COVID-19 infection would have increased, had the survey extended into the 2022 calendar year.

There were discrepancies in vaccine uptake amongst varying political leanings in the study population. This trend was projected before vaccines were even available [20]. It is not well understood why the association between political leanings and vaccine hesitancy exists, but factors such as educational attainment, socioeconomic status, and race/ethnicity are likely implicated [21]. Regardless, the implications are real; between March 1, 2021, and September 1, 2021, an analysis of 3112 counties in the United States showed that a high percentage of Republican voters was associated with lower vaccination rates and higher COVID-19 cases and deaths [21].

This study has several limitations, such as reliance on self-reporting for major endpoints of the study, such as vaccination status. Participants may have had difficulty selecting survey responses if no clear definitions were provided, which was the case for the political leaning question, for example. The cross-sectional design of the study hinders the ability to establish a causal relationship between vaccine hesitancy and the selected demographic factors. Generalizability is also a limitation, as participants from marginalized and minoritized groups were not adequately represented, while White and heterosexual respondents were overrepresented. Potential confounding exists for vaccination rates based on age group as older participants are more likely to have risk factors that may have prompted them to obtain a vaccine sooner than younger participants. This may explain their higher odds of vaccination compared to the younger age groups at the time of the survey. A subsequent analysis is needed to determine if younger age groups approached similar vaccination rates since the study period was also not controlled for when assessing vaccine hesitancy. Moreover, an important temporal context of this study is that six days after the survey release, on October 11, 2021, PSU announced that all employees on the main University Park campus must be vaccinated by December 8, 2021, including students on payroll or graduate assistantships [22].

Using guidance from the 5C model (confidence, complacency, convenience or constrains on convenience, risk calculation and collective responsibility), particularly within the realm of risk calculation, the authors plan to expand upon this study to evaluate what drives vaccine uptake and hesitancy in the PSU population. Specifically, the authors aim to explore how trust in government agencies, trust in elected officials, and methods of obtaining health-related news information (social media versus print newspapers, for example) influence individual decisions to receive a vaccine against COVID-19.

## Supporting information

**S1 File.**
(PDF)

**S1 Checklist. STROBE statement—Checklist of items that should be included in reports of observational studies.**
(DOCX)

## Acknowledgments

The authors want to thank the following individuals and departments for their support with this project: The Penn State University Office of the Executive Vice President and Provost, the interim Dean of the Penn State University College of Medicine; the Senior Vice President and Chief Human Resources Officer for Penn State Health; the Assistant Editor for the University Libraries' Public Relations, the Marketing Department at Penn State University; the Advertising Executive at The Daily Collegian at Penn State; and The Harrell Health Sciences Library at the Penn State College of Medicine.

## Author Contributions

**Conceptualization:** Ryan Murphy, Lauren Pomerantz, Prabhani Kuruppumullage Don, Bradley A. Long.

**Data curation:** Ryan Murphy, Lauren Pomerantz, Prabhani Kuruppumullage Don, Jun Sung Kim.

**Formal analysis:** Prabhani Kuruppumullage Don, Jun Sung Kim.

**Funding acquisition:** Bradley A. Long.

**Investigation:** Ryan Murphy, Lauren Pomerantz, Prabhani Kuruppumullage Don, Bradley A. Long.

**Methodology:** Ryan Murphy, Lauren Pomerantz, Prabhani Kuruppumullage Don, Jun Sung Kim, Bradley A. Long.

**Project administration:** Ryan Murphy, Lauren Pomerantz, Bradley A. Long.

**Resources:** Bradley A. Long.

**Supervision:** Bradley A. Long.

**Validation:** Prabhani Kuruppumullage Don, Bradley A. Long.

**Writing – original draft:** Ryan Murphy, Lauren Pomerantz, Bradley A. Long.

**Writing – review & editing:** Ryan Murphy, Lauren Pomerantz, Prabhani Kuruppumullage Don, Jun Sung Kim, Bradley A. Long.

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
