## [Decision Letter · Decision Letter 0]

19 Jun 2023

PONE-D-23-10814Age and political leaning predict COVID-19 vaccination status at a large, multi-campus, public university in Pennsylvania: a cross-sectional surveyPLOS ONE

Dear Dr. Murphy,

Thank you for submitting your manuscript to PLOS ONE. After careful consideration, we feel that it has merit but does not fully meet PLOS ONE’s publication criteria as it currently stands. Therefore, we invite you to submit a revised version of the manuscript that addresses the points raised during the review process. Please note that revised submission does not guarantee acceptance; It depends on the quality of revision and maybe subject to external peer review again.

We look forward to receiving your revised manuscript.

Kind regards,

Binod Acharya

Academic Editor

PLOS ONE

Journal Requirements:

“*. RM and LP received funding from The Harrell Health Sciences Library at the Penn State College of Medicine to receive compensation as research assistants. The funders had no role in study design, data collection and analysis, decision to publish, or preparation of the manuscript.*”

“RM and LP received funding from the The Harrell Health Sciences Library at the Penn State College of Medicine to receive compensation as research assistants. The funders had no role in study design, data collection and analysis, decision to publish, or preparation of the manuscript.”

Additional Editor Comments (if provided):

• Please thoroughly address the comments raised during the review process.

• Please note that we encourage authors to report confidence intervals (not just the p-values) to convey the uncertainty of the estimates.

• Also, please be reminded that the current presentation of results (in Table 5) is not articulated well enough (both in terms of substance and format requirement of the journal). Importantly, readers might want to see the results of the association between hesitancy and demographic factors of interest when simultaneously controlling for all of them (i.e. age and political leaning and maybe more covariates about which you have collected the data). Also, note that some of the numbers presented do raise eyebrows. For example, the ORs comparing far-light versus far-left is as small as 0.06 (the back-of-the-envelope calculation of unadjusted ORs gives a very different number). Please also attach the associated R code during the revised submission. 

• Moreover, please be reminded that we do not do copy editing and it’s your responsibility to address any and all typos. A few examples: “Conversative” for conservative; “R™” with unnecessary trademark, and “COVID” without 19.

• Specific comments on the text. Please note that this sentence below gives an impression of as if Table 1,3,4 present associations. They don’t. Also redundant Table 1. And you didn't "made" associations.

“Associations between selected factors such as gender (Table 1), age (Table 1), political orientation (Table 3) and vaccine hesitancy (Table 4) were made with Chi-square tests and generalized linear regression models on R™ version 4.2.1 (2022-06-23).”

• In the abstract: there appears to be a typo on OR (4.26 vs 4.30). Again, I note that the associated P with (big) OR >4 is 0.04. Please report the width of CI as well. Also, given the specific study sample is the PSU, what exactly do authors have in mind when you say “public health efforts such as vaccine campaigns and policy decisions” in your conclusion.

Reviewers' comments:

Reviewer's Responses to Questions

**Comments to the Author**

1. Is the manuscript technically sound, and do the data support the conclusions?

Reviewer #1: Yes

Reviewer #2: Yes

2. Has the statistical analysis been performed appropriately and rigorously? 

Reviewer #1: No

Reviewer #2: Yes

3. Have the authors made all data underlying the findings in their manuscript fully available?

Reviewer #1: Yes

Reviewer #2: Yes

4. Is the manuscript presented in an intelligible fashion and written in standard English?

Reviewer #1: Yes

Reviewer #2: Yes

5. Review Comments to the Author

Reviewer #1: This study examines factors associated with COVID-19 vaccine hesitancy at Pennsylvania State University (PSU). The study utilized a REDCap™ survey distributed to students, faculty, and staff between October and November 2021, and received 4,231 responses. The authors found that age and political leaning were associated with vaccine hesitancy, with older individuals and those who identified as liberal being more likely to report COVID-19 vaccination. The authors note that the survey was distributed prior to the emergence of the Omicron variant and that only 80.3% of participants had never tested positive for COVID-19, which may have influenced participants' perceived need for vaccination.

In addition, some issues in this manuscript either need to be addressed or acknowledged in the limitations before it is considered for publication.

First, the manuscript does not provide a literature review on the association between vaccine incentive programs and vaccine uptake in the United States. The authors should include a comprehensive review of relevant studies in the introduction and discussion sections. Please refer to the following two articles.

https://jamanetwork.com/journals/jamanetworkopen/article-abstract/2786986

https://academic.oup.com/aje/article/192/4/510/6987267

Second, the authors did not report the response rate of the survey, which is important to assess the representativeness of the sample.

Third, the study design is cross-sectional, which makes it difficult to establish a causal relationship between the factors investigated and vaccine hesitancy. In addition, the authors did not provide a detailed description of the recruitment methods, which makes it difficult to assess the potential for selection bias. The authors should acknowledge these limitations and discuss how they may have affected the results and the generalizability of the findings.

Fourth, the authors did not control for other potential confounders that may be associated with vaccine hesitancy, such as education level, race/ethnicity, and health status. Was there any effort to measure or control for the influence of social media and other external sources of information on vaccine hesitancy or to assess participants' trust in data from various sources?

Fifth, the manuscript lacks a clear discussion of the practical implications of the findings for public health policy and practice. The authors should provide a detailed discussion of the implications of their findings for vaccine campaigns and policy decisions, and they should also discuss the potential for future research in this area.

In conclusion, while this manuscript provides some insights into the factors associated with vaccine hesitancy and uptake among members of the PSU community, there are several shortcomings that need to be addressed before publication. Specifically, the authors need to cite recent literature on vaccine hesitancy and measures (for example, vaccine incentive program) to break this hesitancy, provide a more detailed description of the study design, recruitment methods, statistical models and limitations of the study. They also need to address potential sources of bias and confounding.

Reviewer #2: This journal article examines the impact of vaccine hesitancy on higher education institutions during the COVID-19 pandemic. The objective is to understand the factors associated with vaccine hesitancy and uptake in order to inform policies and provide reliable information during public health emergencies. The study focuses on a large, multi-campus public university in Pennsylvania and analyzes the relationship between age, gender, political leaning, and COVID-19 vaccination status. Data was collected through a survey distributed to students, faculty, and staff between October 5 and November 30, 2021, resulting in 4,231 responses. Statistical analysis using Chi-square tests and logistic regression models revealed that age and political leaning significantly influenced vaccine hesitancy. Specifically, individuals aged 56 years or older were more likely to be vaccinated compared to those aged 18 to 20, and liberal individuals had higher vaccination rates compared to far-left individuals. These findings highlight the importance of considering age and political leaning when designing vaccine campaigns and making policy decisions to improve vaccine uptake in university communities.

Please note that following comments are provided to support the peer-review process and should be taken into consideration for the improvement of the article.

The article lacks clarity regarding the methodology employed to identify the political preferences of the respondents. The authors should provide a detailed explanation of the approach utilized to collect and categorize political leaning information.

It would be beneficial to include information on how the potential bias or subjectivity in assessing political preference was minimized or accounted for in the analysis.

The authors should consider providing additional information about the political spectrum utilized in the study to help readers understand the categories and their definitions.

The article would benefit from a more explicit discussion of the specific concerns related to the Johnson & Johnson vaccine, such as the reported cases of blood clotting and the subsequent FDA advisory. Providing a clear overview of these concerns and their potential impact on vaccine hesitancy would enhance the readers' understanding.

It is important to explore and discuss any differences in vaccine hesitancy rates or perceptions between the Johnson & Johnson vaccine and the Pfizer and Moderna vaccines. Comparing the hesitancy levels across the three vaccines would provide valuable insights into how specific safety concerns may influence individuals' decision-making.

6. PLOS authors have the option to publish the peer review history of their article (what does this mean?). If published, this will include your full peer review and any attached files.

Reviewer #1: No

Reviewer #2: No

---

## [Author Response · Author response to Decision Letter 0]

31 Aug 2023

The "Response to Reviewers" document is included as a file. Thank you!

---

## [Editor Report · Decision Letter 1]

10 Sep 2023

Age and political leaning predict COVID-19 vaccination status at a large, multi-campus, public university in Pennsylvania: a cross-sectional survey

PONE-D-23-10814R1

Dear Dr. Murphy,

We’re pleased to inform you that your manuscript has been judged scientifically suitable for publication and will be formally accepted for publication once it meets all outstanding technical requirements.

Kind regards,

Binod Acharya

Academic Editor

PLOS ONE
---

## [Editor Report · Acceptance letter]

12 Sep 2023

PONE-D-23-10814R1 

Age and political leaning predict COVID-19 vaccination status at a large, multi-campus, public university in Pennsylvania: a cross-sectional survey 

Dear Dr. Murphy:

I'm pleased to inform you that your manuscript has been deemed suitable for publication in PLOS ONE. Congratulations! Your manuscript is now with our production department. 

Kind regards, 

on behalf of

Mr. Binod Acharya 

Academic Editor

PLOS ONE